

# Flood Impacts on a Water Distribution Network

Chiara Arrighi[1], Fabio Tarani[2], Enrico Vicario[2], and Fabio Castelli[1]

[1]Università di Firenze, DICEA (Department of Civil and Environmental Engineering)
[2]Università di Firenze, DINFO (Department of Information Engineering)

*Correspondence to:* Chiara Arrighi (chiara.arrighi@dicea.unifi.it)

**Abstract.** Floods cause damage to people, buildings and infrastructures. Due to their usual location near rivers, water utilities are particularly exposed; in case of flood, the inundation of the facility can damage equipment and cause power outages. Such impact lead to costly repairs, disruptions of service, hazardous situations for personnel and public health advisories. In this work, we present an

analysis of direct and indirect damages of a drinking water supply system considering the hazard of a riverine flooding as well as the exposure and vulnerability of the system components (i.e. pipes, junctions, lifting stations etc.). The method is based on the combination of a flood model and an EPANET-based piping network model implementing Pressure-Driven Demand, which is more appropriate when modeling water distribution networks with many off-line nodes. The two models

are linked by a semi-automated GIS procedure. The evaluation of flood impact on the aqueduct network is carried out for flood scenarios with assigned recurrence intervals. Vulnerable elements exposed to the flood are identified and analyzed in order to determine their residual functionality and simulate failure scenarios. Impact metrics are defined to measure service outage and potential pipe contamination. The method is applied to the water supply system of the city of Florence (Italy),

serving approximately 385 000 inhabitants. Results show that for the worst failure scenario 420 km of pipeworks would require flushing and disinfection with an estimated cost of 21 Mio €, which is about 0.5% of the direct flood losses evaluated for buildings and contents.

## 1 Introduction

Extreme weather events and major natural disasters are listed in the top five global risks, in terms

of both likelihood and impact (World Economic Forum, 2017). Climate change perspectives (IPCC, 2013; Lung et al., 2013) raise additional concerns about floods due to their consequences on population (Ashley and Ashley, 2008), environment (Christodoulou, 2011), urban areas and infrastructures





(Meyer et al., 2013; Emanuelsson et al., 2014; Khan et al., 2015; Short et al., 2012). This leads to an increasing interest in studying flood impact, as shown, for instance, by the sustainability criterion

adopted for flood risk mitigation strategies in EU countries (EU Parliament, 2007), which promotes quantitative flood risk assessment (Merz et al., 2010) and flood damage maps (de Moel et al., 2009; Meyer et al., 2013).

Flood damage on structures and infrastructures is classified into direct and indirect, the former being caused by physical contact with floodwater and the latter occurring far from the event, either

in space or time (Thieken et al., 2006). On the one hand, direct losses to private dwellings, household contents and economic activities can be estimated through damage curves, which relate water depth to relative losses (Smith, 1994); on the other hand, interdependence of assets in network infrastructures potentially induces impacts outside flooded areas, sometimes with substantial effects (Gil and Steinbach, 2008). Hence, the assessment of flood impact on networks in which some com-

ponents come into direct contact with water requires the evaluation of the potential consequences on the overall system behaviour. As a matter of fact, failure of crucial infrastructures may lead to cascade events and trigger technological disasters (Cruz et al., 2004). Cascading events are more likely to occur during a natural disaster than during normal plant operation because of the increased chance of multiple, simultaneous failures. While flood damage evaluation to buildings and their con-

tents is becoming increasingly available (Merz et al., 2010), the quantification of direct and indirect impacts on critical infrastructures is less common (Lhomme et al., 2013; Michielsen et al., 2016; Emanuelsson et al., 2014).

Among safety critical infrastructures are fresh water supply systems (WSS, see Table 1 for a list of acronyms used in the paper) and water treatment plants, which can be severely affected by floods

since they rely on electric power, mechanic and electronic devices. Water supply and sanitation is widely considered as a main factor in environmental sustainability, human health, social services and resilience (WHO, 2011; Luh et al., 2017). In particular, water distribution networks are complex systems composed by a number of subsystems in charge of abstraction from the source, transportation, treatment and distribution. The assessment of flood risk requires the evaluation of the three risk

components – i.e. hazard, vulnerability and exposure – for each subsystem and the assessment of functional dependencies (Serre et al., 2011). In particular, flood *hazard* of a component is the probability of being flooded, which can be evaluated through flood maps; *exposure* is the position with respect to inundation extent and *vulnerability* is the attitude of being harmed (Meyer and Messner, 2005). Vulnerability of a WSS can be intended as the susceptibility of a single network portion (e.g.

a valve) as well as the fragility of the whole system (e.g. the water distribution) due to the failure of a system component. This distinction is particularly crucial for network infrastructures where the failure of one node may trigger harmful effects also geographically far from the affected area, constituting an indirect damage.



**Table 1.** Acronyms used in the paper

| Acronym | Definition |
|---------|------------|
| DWTP | Domestic Water Treatment Plant |
| PDD | Pressure-Driven Demand |
| RI | Recurrence Interval |
| WDN | Water Distribution Network |
| WSS | Water Supply System |

Vulnerable WSS components are often located in low-lying areas or nearby rivers, with a conse-
quent high exposure to inundations. Flood events affecting water utilities can lead to costly repairs,
disruptions of service and public health advisories (U.S. Environmental Protection Agency, 2014).

The management of flood risk entails a combined approach comprising mitigation, preparedness,
response and recovery (WHO, 2011). Among the mitigation activities, the identification of hazard
and a comprehensive vulnerability analysis are recognized as pre-eminent. Risk assessment is a
fundamental support for decision makers because it increases the awareness and fosters the adoption
of mitigation strategies (Large et al., 2014).

The implementation of Water Safety Plan promoted by the World Health Organization (WHO)
and International Water Association (IWA) (Bartram et al., 2009) aims at harmonising hazard and
risk assessment procedures through an appropriate method. It identifies issues on treatment plants
and source water quality (Ginandjar et al., 2015) as the main hazards associated with floods. Floods
and heavy rainfall are associated with elevated turbidity and dissolved organic matter (Göransson
et al., 2013; Murshed et al., 2014), which can affect drinking water purification whose source is
a surface water body or storage reservoir. However, if indirect/cascade effects are accounted for,
other impacts should be considered such as those related to power outage, which is likely to occur
if electric devices, e.g. valves and lifting stations, are affected (Khan et al., 2015). In fact, a short-
term loss of the electric power may induce pressure fluctuations or intermittent supply, which may
lead to ingress of contamination from leakage orifices and air vacuum valves (Ebacher et al., 2010).
Thus, besides the economic costs the contamination may cause, there are repercussions on social
and operational domains characterizing urban water systems (Blackmore and Plant, 2008; Hrudey
et al., 2006). Hence, a comprehensive flood risk assessment of WSS's should integrate a flood model
and a WSS model capable of properly representing the network behavior in low pressure conditions
(Seyoum and Tanyimboh, 2016).

This work aims at implementing a method to evaluate flood impact on a WSS accounting for
both direct and indirect damage on technological systems and inhabitants. Hazard, vulnerability and
exposure of system components are assessed through a semi-automated procedure integrating the
GIS representation of flood scenarios with an hydraulic network model with Pressure-Driven De-





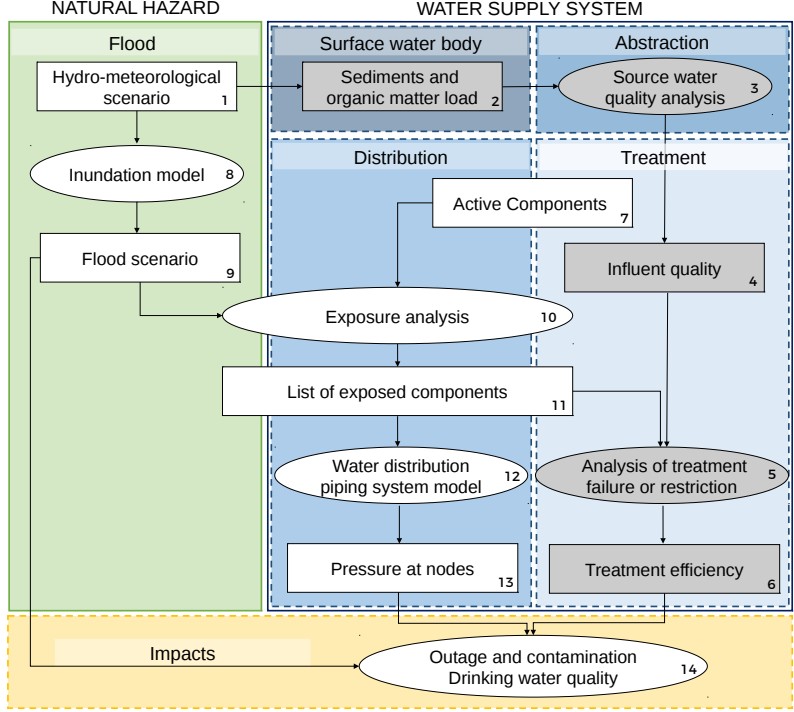

**Figure 1.** Flood risk assessment scheme for WSS (ellipses stand for activities, rectangles represent data flow; Shaded boxes represent activities that are not carried out in this work).

mand (PDD). Failure scenarios are based on the analysis of exposure and vulnerability of critical network components, e.g. lifting stations. Three metrics for the assessment of flood impact are introduced and the model is tested on a case study.

## 2   Materials and Methods

The flood risk assessment of a WSS requires a comprehensive approach including several scales of analysis and models in order to capture the dependencies between environmental forcing (i.e. a natural hazard) and WSS components besides the inner dependencies of the WSS. Impacts on the WSS pertain to quantity and quality of the drinking water. Figure 1 depicts the logic flow to estimate flood impacts on each component of the WSS considering a configuration with a surface water body source, e.g. a river.

Reading the scheme clockwise, at catchment scale the hydro-meteorological event (1 on the diagram) bears turbidity, due to the high concentration of suspended sediments, and organic matter load, which affect the surface water body (2). When reaching the abstraction, the quality of source



**Table 2.** Main impacts associated with flooding for WSS based on surface water source

| WSS component | Direct flood impact | Consequence |
|---|---|---|
| Abstraction | turbidity | abstraction interruption |
| | organic matter load | restriction of treatment |
| Treatment | power shutdown | loss or restriction of treatment works |
| | instrumentation failure | loss of control |
| | drinking water contact with floodwater | contamination |
| Distribution | power shutdown | pressure fluctuations |
| | | intermittent supply |
| | | contamination |

water needs to be analyzed (3) to determine if the influent (i) is suitable for a standard treatment,
(ii) requires adjustments of the treatment process, (iii) is not appropriate, leading to a temporary
interruption of abstraction (4). Uncontrolled or special source water quality may directly affect the
treatment with possible failures of the process sections and consequences on treatment efficiency
(5,6). Treatment plants are also susceptible of failure or restrictions if active components are flooded.
Active components (7) are those powered by electricity such as valves, pumps, chemical dosers etc.
Also the WDN, which relies on elements sensitive to power outage (e.g. lifting stations), is affected.
The inundation model (8) generates a flood scenario (9), i.e. an inundation map which allows to
identify exposed objects. Exposure analysis (10) produces a list of exposed components (11), both
for WDN and treatment, whose possible failure should be simulated in a piping distribution system
and a treatment plant models respectively (12). Therefore, the results of the models in terms of pres-
sure at nodes (13) and treatment efficiency allows for evaluating the impacts on water quantity – i.e.
outages due to pressure fluctuations or intermittent supply – and quality – e.g. risk of contamination
(14). The main consequences of flood impacts are summarized in Table 2.

This work focuses on the evaluation through a numerical model of flood impacts on the WDN,
shown in the central panel of Figure 1. The model is composed by two main sub-models: the inun-
dation model, which is used to estimate flood parameters for an assigned probability scenario (e.g. a
flood map) and the WSS model.

### 2.1 Inundation model and exposure analysis

The inundation model exploits the input flow hydrograph calculated for a hydro-meteorological sce-
nario to produce a raster map showing the representative flood parameters, in particular water depth.
Computation can be performed using simplifications of the Navier-Stokes equations and adopting
several numerical schemes and spatial resolution of the computational domain (Hunter et al., 2008).
Accurate forecast of flood propagation in urban environments usually requires 2D models with an



adequate description of the street/building pattern, leading to a desired resolution of about 1 m for the
computational grid (Apel et al., 2009). In this context, the increasing availability of geographic data
such as LiDAR-derived Digital Terrain Models (DTM) and anthropic structures and infrastructures
ease the set-up of the model. Nevertheless, some issues such as needed computational effort and the
definition of representative roughness coefficients still remain. As an alternative, parsimonious hy-
draulic models are also accepted as a compromise between accuracy and computational effort when
steady state approximations and large and cumbersome computational domains are not sustainable
(Apel et al., 2009; Arrighi et al., 2013).

The hydraulic model adopted is composed of two parts. Firstly, the river is represented with a
1D unsteady flow model and the urban flood-prone area is modelled as a system of interconnected
quasi-2D storage cells. A Digital Surface Model (DSM) of 1 m resolution and 0.25 m vertical ac-
curacy derived from LiDAR surveys is used for the detailed representation of the flow domain at
streets/buildings scale. Buildings are, by default, considered as waterproof blocks. The computation
of flood propagation is performed through an implicit 1D finite-difference scheme of the general
equation of unsteady flow (i.e. continuity and momentum conservation equations). The quasi-2D
hydraulic model for the floodplain consists of several storage areas (cells) connected to the river
banks through a set of lateral weirs, whose geometry is extracted from topographic survey. When
the inundation starts, the quasi-2D module – governed by continuity and stage-storage relations –
calculates water levels from the volume stored in the cell. Flow between adjacent cells is described
by a weir equation accounting for backwater effects. The details of the model construct and equa-
tions adopted in the HEC-RAS framework (for both 1D and quasi-2D modules) are described in
Arrighi et al. (2013). Moreover, the availability of inundation maps from local and national water
authorities is growing due to the evolving normative frameworks in flood risk management (EU Par-
liament, 2007). Thus, official inundation maps can be adopted if accessible and adequate in spatial
resolution in the area of interest. Geographic Information Systems (GIS) are unavoidable to identify
exposed asset. All the components of the WDN, both active and passive must be geo-referenced to
be compared with inundation maps for assigned scenarios.

Exposure analysis consists of four steps. Firstly, the coordinates of the WSS point components
(nodes, reservoirs, lifting stations, etc.) are exported from the WSS model to a layer in the GIS
environment, so that a new vector is created whose coordinate reference system is assigned in the
shapefile properties. Afterwards, the raster inundation map is imported into the GIS workspace and
converted if necessary to a compatible reference system. The raster cell information (i.e. water depth)
is then extracted over the point feature and added as attribute using open plugins (e.g. Point Sampling
Tool available for QGIS). The point features whose water depth attribute is larger than a threshold are
considered exposed and added to the list of exposed asset. The threshold, in this work, is assigned a
fixed value of 0.25 m. For each component pertaining to exposed asset and failure-prone, local water
depth is compared to a threshold depth which takes into account local geometry and functional



dependencies. If calculated depth exceeds the threshold, the component is marked as failed and its
properties modified in the WSS model (section 2.2) to reproduce the failed configuration.

## 2.2 Distribution Network Model

The model is based on the freely available EPANET libraries, which calculate time-varying pressures
at the nodes given a set of initial tank levels, pump switching criteria, node base demands and demand
patterns. In particular, EPANET can be launched by other software through a set of DLL libraries.

One drawback of the standard EPANET implementation is its strict demand-driven approach,
which stems from the primary goal of simulating correctly operated networks. In such networks,
pressure at each node is sufficient so as to allow withdrawal of required demand from each node, so
that demands can be assumed as defined input data. However, when simulating strongly off-design
networks, nodes featuring a reduced pressure are possible and quite common, so that a pressure-
driven approach is needed (Cheung et al., 2005; Walski et al., 2017). PDD models differ from con-
ventional ones in that demands at nodes are not attributed a priori, but their value depends on the
current pressure at the considered node.

In particular, and consistently with practice, the model assumes that each node can be in one of
three states:

**fully served**  - if $H_i \geq H_{\text{service}}$, the node is able to withdraw its nominal demand;

**partially served**  - if $H_{\text{service}} > H_i > 0$, the node withdraws a reduced demand proportional which
can be expressed as

$$D_i = D_{\text{nom},i} \left( \frac{H_i}{H_{\text{service}}} \right)^{\alpha} \tag{1}$$

where $\alpha$ is a constant exponent set to 0.5.

**non-served**  - if P = 0, the node is unable to withdraw any water, yielding to null demand.

EPANET allows two types of nodes: *nodes* are assigned a time-varying, pressure-independent
demand, and can be effectively used to model fully served users, whereas *emitters*, conceived
to model fixed cross-section water outlets such as fire hoses and orifices, adequately model the
aforementioned behaviour of partially served users. Emitters are defined by a fixed exponent $\alpha$, equal
for all instances, and a flow coefficient which represents the volume flow rate for unitary pressure
loss across the orifice. Unfortunately, emitters do not cope well with calculated negative pressures,
attributing a negative – i.e. entering – flow rate where such negative pressures occur.
In order to cope with this issue, a MATLAB code has been implemented so as to run transient
simulation while correctly using a PDD approach. The code – as shown in Figure 2 – works as
follows: three node states are defined: "2" for served nodes, "1" for partially served ones and "0"
for non-served ones; type 2 and type 0 nodes are modelled as EPANET nodes with nominal demand




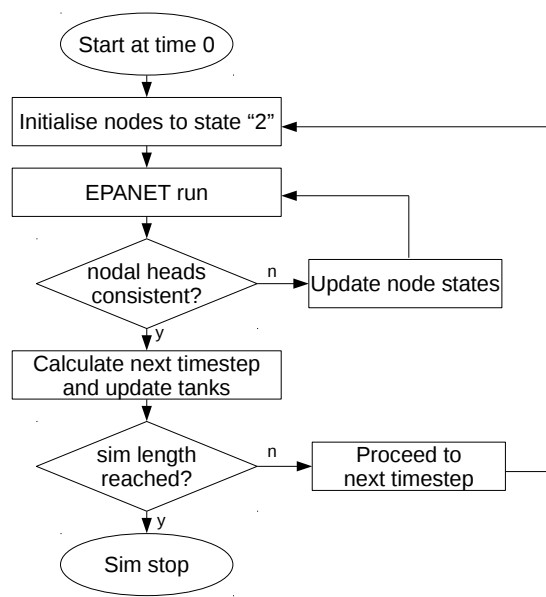

**Figure 2.** Diagram of PDD model implementation.

equal to the assigned nominal demand $D_i$ and zero respectively, whereas type 1 nodes are modelled
as emitters whose flow coefficients are calculated to ensure that $D_i = D_{\mathrm{nom},i}$ if $H_i = H_{\mathrm{service}}$.

For each timestep, a first trial simulation is run with all nodes in state 2 in order to get the expected pressures. Afterwards, each node is checked to assess whether its pressure is in the pressure range corresponding to the current flow regimen and, if this is not the case, its state is accordingly raised or lowered by one unit (namely, it is not possible to jump from state 2 to state 0 and vice versa). After node states have been changed, simulation is repeated till no more node state change is necessary. Calculated flow rates and pressures are considered to represent network operation during the following timestep. In particular, flow rates are used to calculate the time to the next event (tank being filled or emptied), and the first event affecting network topology is considered (e.g. demand change, or pump setting toggle due to time pattern, tank getting empty or full). Tank levels are the updated and simulation proceeds to the next timestep.

The described procedure allows to calculate pressure and supplied demand at each node for each timestep, therefore fully estimating the network state in each moment.

### 2.2.1 Model initialisation

The model, featuring non memory-less elements (tanks), needs to be correctly initialised. In normal operation, tank levels undergo a daily pattern of filling and emptying, according to demands and





water availability. In order to appropriately initialise tank levels, a warm-up simulation is run by randomly initialising tank levels and checking their value every 24 hours. If the calculated levels with a time difference of 24 hours differ by less than a tolerance parameter, the model is considered to be in long-term steady state and water level for each tank and time value are saved in a matrix, 215 which can be thereafter be used to initialise such values for the forthcoming simulations.

## 2.3 Definition of metrics

In order to evaluate the global impact of the flood on network operativeness and integrity, two indicators have been defined. In the first place, impact of the flood on network operation is assessed through evaluation of the number of inhabitants experiencing lack of service. To this aim, data about 220 population density in the area made available by the Italian National Institute of Statistics (ISTAT, 2011) are used. Such data define 2186 polygonal zones in the studied area with areas ranging from 156 to $2.48 \times 10^6\,\mathrm{m}^2$) and give the population for each of such zones.

Inhabitants are assigned to nodes as follows: an uniform demand per capita is assumed in each area and calculated, and the number of inhabitants for nodes pertaining to that area is estimated 225 accordingly. In particular:

$$P_{i\in A} = \frac{D_{\mathrm{nom},i}}{\sum\limits_{k\in A} D_{\mathrm{nom},k}} P_A \tag{2}$$

where $P_i$ is the population assigned to node $i$ belonging to area $A$ and $P_A$ is the total population of area $A$. The global damage parameter Non-Served Population (NSP) is therefore estimated as the sum of population attributed to nodes with reduced or null pressure, i.e.

$$230 \quad \mathrm{NSP} = \sum_{i\in I} P_i \quad \text{with } I = \{i \mid H_i < H_{\mathrm{service}}\} \tag{3}$$

where $H_{\mathrm{service}}$ is the minimum head required to consider a node fully served ($5.0\,\mathrm{m}$ in the case study).

As a second parameter, network damage due to pipe contamination is evaluated by calculating the total length of pipework to be decontaminated. A pipe is considered to be contaminated if at any 235 point in time the head inside the pipe is lower than the flood water head outside or below zero, i.e.

$$L = \sum_{i\in I} \sum_{j\in J_i} L_j \quad \text{with } I = \{i \mid H_i < \max(H_{\mathrm{flood}}, 0)\} \tag{4}$$

where $J_i$ is the set of pipes with either end connected to node $i$.

## 3 Case study

### 3.1 Flood scenarios

240 The studied area is the municipality of Florence, Italy, with an areal extent of $102\,\mathrm{km}^2$. The area hosts $383\,000$ inhabitants, with the highest population density concentrated in the city centre along





the river banks. Documents witness that the town has a long account of floods since the Middle Ages, as confirmed by more recent hydrologic-hydraulic studies (Caporali et al., 2005), which show that it is prone to floods also for low recurrence interval (30-year return period). For such frequent events, only the lower-lying suburbs are affected (brown areas on the center left side of Figure 10), whereas more severe scenarios (recurrence interval of over 200 years) affect the whole city including the historic area. Flood risk in the area studied is estimated in 55 M€/year if only direct tangible losses to buildings, household contents and commercial activities are taken into account (Arrighi et al., 2016). In this context, the analysis of flood risk to the WSS is crucial to better understand the potential adverse consequences on strategic infrastructures and estimate the recovery costs.

The meter scale DTM used for the hydraulic model is freely available in the regional cartographic repository (dati.toscana.it/dataset/lidar). The hydraulic data (hydrographs and river water profiles) are made available by the catchment authority ("Autorità di Bacino del Fiume Arno"), which is in charge of flood risk management and water resource planning.

For the application of the method described in Section 2, four flood scenarios with given recurrence interval RI are considered: a frequent scenario (i.e. 30-year return period), two medium recurrence intervals (100 and 200 years) and a rare scenario (500 years). Accordingly, four inundation raster maps are generated for during the exposure analysis.

### 3.2 Water distribution network

The studied WSS features one main treatment facility, 17 tanks and the pipework to supply drinking water for domestic and industrial use.

Fresh water supply is ensured by the river Arno, which flows westbound amidst the urban area. Water is abstracted from the river by three 373-kW pumps in the treatment plant "Anconella", which is located in the left bank and designed to process $4\,m^3$/s (Fig. 3). The water undergoes treatment and reaches the lifting station, where six 710-kW pumps ensure a maximum head of 60 m and feed the distribution network. The storage tanks are mostly located at high altitudes and feature a total operative volume of $48\,620\,m^3$.

An EPANET model of the WSS is provided by the utility operator Publiacqua SpA. The model is barely skeletonised, and consists of 4863 nodes and 12 436 pipes for a total length of the modeled piping network of 619 km.

The WSS elements most vulnerable to floods are the lifting stations and the pumps feeding the storage tanks, because they rely on electrical power and are affected by power shutdown. The WSS vulnerable components exported to the GIS environment have an attribute of elevation which is compared to the flood depth for each flood scenario. If the flood depth above the vulnerable component exceeds 0.5 m, the failure of the component is assumed and the object is listed to be switched off in the water distribution piping system model.





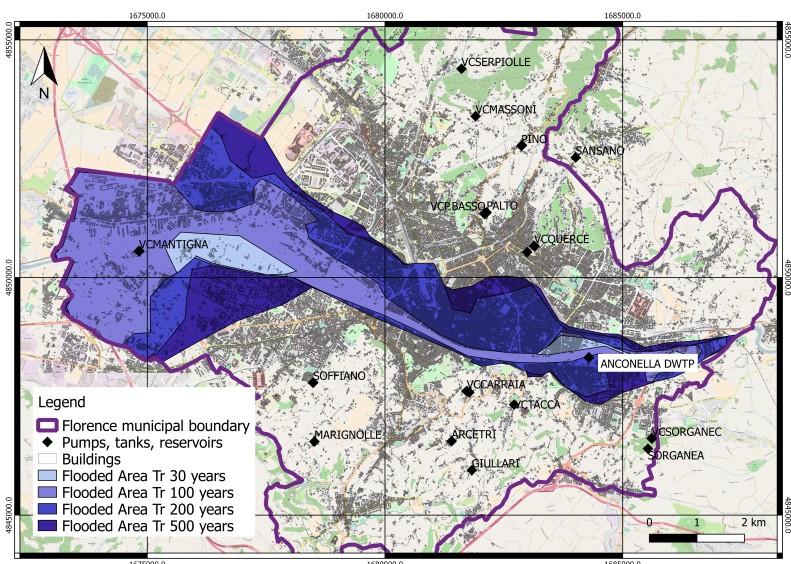

**Figure 3.** Flooded area for the four recurrence intervals and exposure of vulnerable components. Reference coordinate system is EPSG:3003.

## 4 Results

### 4.1 Flood scenarios and exposure analysis

Figure 3 shows the results of hazard analysis. For the 30-year RI an area of 2.5 km$^2$ is flooded, with

an average water depth of 1 m. For the 100-year recurrence interval the flooded area increases to 12.7 km$^2$ with an average flood depth of about 1 m. For higher RI (200 and 500 years), the affected areas rise to 20 and 27 km$^2$ and average depths to 1.2 and 1.7 m respectively.

Table 3 reports the results of the exposure analysis. For RI= 30 years, none of the vulnerable WSS components is affected. For RI=100 years, the tank labelled "VCMantigna" is exposed to flood, yet

the average flood depth in a buffer zone of 25 m radius is about 0.15 m, thus the electrical devices are considered functional. For RI=200 and 500 years the drinking water treatment plant (DWTP) at Anconella, shown in the right-hand side of Figure 3, is flooded with a water depth exceeding 0.85 m. For these scenarios, issues are expected because of drinking water treatment restrictions, loss of control and power shut-down of the lifting station. The "VCMantigna" tank is still exposed, with

water depths as high as 2 m for RI=500.

### 4.2 Failure scenarios

Results relating to the 200-year and 500-years recurrence intervals are reported, i.e. those for which failure of the DWTP is envisaged. In particular, two scenarios are considered: in scenario 1, it is





**Table 3.** Summary of exposed components

| Recurrence interval years | Inundated Area km$^2$ | Depth at lifting station m | Depth at "VCMantigna" m |
|---|---|---|---|
| 30 | 2.5 | - | - |
| 100 | 12.7 | 0.85 | 0.15 |
| 200 | 20.5 | 0.90 | 0.73 |
| 500 | 27.8 | 1.50 | 2.00 |

assumed that the DWTP completely stops providing fresh water to the system. Whereas in scenario 2
some backup system is envisaged to keep one of the three main pumps feeding the network in
operation.

#### 4.2.1 Affected areas

The heads at nodes after 120 minutes from the lifting station failures are shown in Figure 4 in failure
scenarios 1 (panel a) and 2 (panel b). After 120 minutes from the shutdown in the failure scenario 1
(Figure 4, panel a), about 50% nodes already experience heads lower that 1 m, where just three zones,
one in the westernmost part of the network (due to the lower altitude favouring piezometric head) and
two on the northern and southern hills (due to local tanks providing capacity, see Sect. 4.3) feature
heads higher than 20 m. After six hours (Fig. 5), the number of served nodes is further reduced, with
only the westernmost part of the network and southern hills (low altitude and higher with a great
number of tanks respectively) being served.

For what concerns failure scenario 2 (Figure 4, panel b), most nodes of the network are operational
after 120 minutes from the shutdown, with pressures in the minimum range of residual level of
service(1-10 m). A few nodes on the northern hills (about 15%) experience heads lower than 1 m
and a significant part of the western city in the right bank experience heads comprised between 10
and 200 m due to their low terrain elevation.

In both cases the service disruption, due to insufficient pressures, affects also nodes outside the
inundated area, thus it can be considered an indirect impact of the flood triggered by the failure of
the lifting station due to the physical contact with water.

#### 4.2.2 Calculated metrics

The time evolution of aggregate service metrics are calculated for the two aforementioned failure
scenarios. Population Not Served is shown in Figures 5 and 6 as a fraction of total population for the
two scenarios.

In failure scenario 1, the complete shutdown of the DTWP pumping station puts almost 50%
of the population in a no service condition after 3 hours, consistently with the dynamics shown in




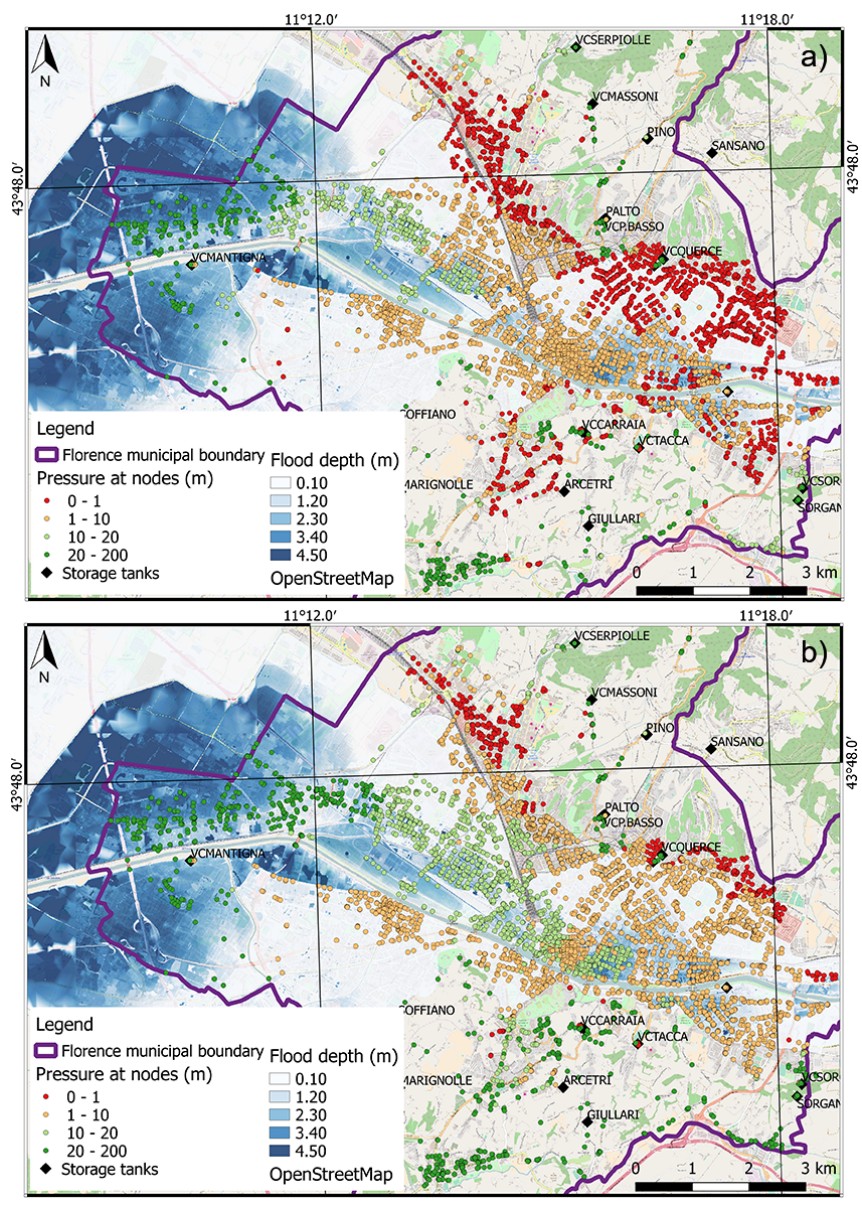

**Figure 4.** Inundation map for the 500-year recurrence interval and pressure at nodes 120 minutes after lifting station failure for scenario 1 (no pumps on) (a) and scenario 2 (one pump on) (b). Reference coordinate system is EPSG:WGS84.

Sect. 4.2.1. After about six hours, such condition extends to the 70%. If inhabitants experiencing insufficient pressure are considered, total affected population is about 90%.




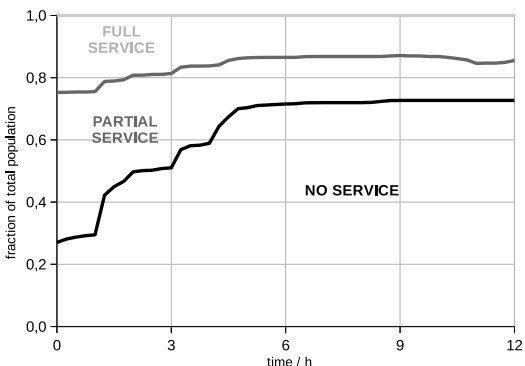

**Figure 5.** PNS as a fraction of total population in failure scenario 1

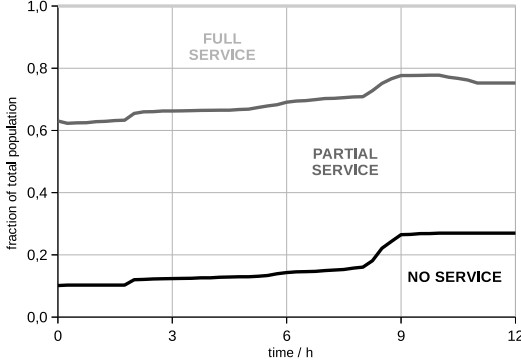

**Figure 6.** PNS as a fraction of total population in failure scenario 2

In failure scenario 2, total affected population ranges from 62 to 77%. Nevertheless, inhabitant experiencing no service at all are about 15%, rising to less than 30% only after 9 h.

For what concerns evaluation of network damage, Figures 7 and 8 show the length of contaminated
pipe as a function of time for the two studied failure scenarios.

Again, scenario 1 shows a critical situation, where about 25% of the network undergoes contamination risk shortly after the shutdown and 68% of the network is out of service just six hours afterwards. In scenario 2, the contaminated pipe length fraction rises from 9% to 26% in the first 12 hours, thus suggesting a milder impact. Nevertheless, caution must be paid for the risk of back-
flow towards nodes which lie on the borders of the served areas.

In principle, contaminated pipe length does depend of the RI considered, since higher floodwater depth leads to higher contamination risk. Nevertheless, results show that in the studied case there is little difference between the 200 and 500-year recurrence intervals.




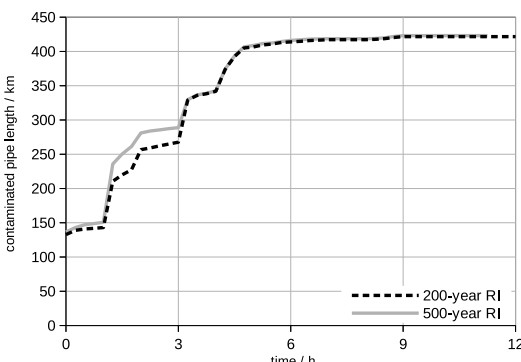

**Figure 7.** Contaminated pipe length in failure scenario 1

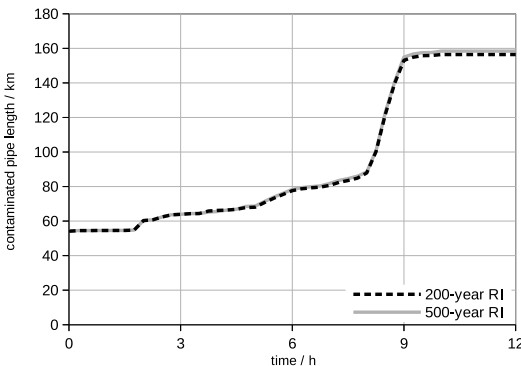

**Figure 8.** Contaminated pipe length in failure scenario 2

If a pipe has been contaminated, i.e. it has potentially received water from the ground, it need

to be disinfected before being put in service again. Disinfection is usually achieved by flushing: trailer-mounted equipment pumps a disinfecting solution (e.g. liquid chlorine or sodium hypochlorite) through a closed piping loop. Firstly, service laterals are closed and customers are connected to bypass piping. Subsequently, the cleaning solution is pumped from a tank on the equipment trailer through the length of the pipe to be cleaned. After cleaning, the solution is neutralized and pumped

to a sanitary sewer. The entire system is then flushed (including laterals) to eliminate sediments and completely remove the disinfecting fluid. From the operational point of view, discharge is monitored during the flushing to assure a sufficient contact time. Chlorine residuals after disinfection are recorded to meet the sanitary standards. An average cost, which can be considered representative of the order of magnitude of the disinfection-flushing operation is 50 € per meter of cleaned pipe

(Ellison et al., 2003); nevertheless, during emergencies, costs may increase due to the dispropor-




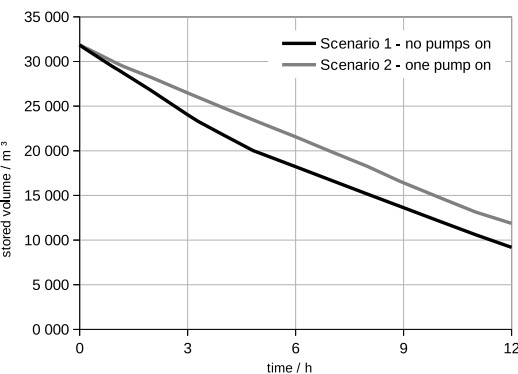

**Figure 9.** Volume stored in tanks as a function of time since failure

tion between available and needed resources. According to calculated values of contaminated pipe length, flood damage to the WDN can be estimated in 21 and 8 M€ for failure scenarios 1 and 2 respectively, which correspond to approximately 0.5 and 0.2% of the direct losses to buildings and contents for the same flood scenario estimated in Arrighi et al. (2016).

### 4.2.3 Tank dynamics

Figure 9, shows the water volume stored in the tank system at a given time after the failure. In scenario 1, where no water is provided by the DWTP, the entire demand is met by withdrawing water from the tank system. This is highlighted by the average slope of the curve in the first three hours (about 0.75 $m^3/s$, which corresponds to half of the total demand in normal conditions. After about 3 and 5 h, reservoir configuration changes, so that the average slope of the volume of the tanks changes.

Slope changes in both curves are caused both by demand variations and tanks becoming empty. In particular, the abrupt change for failure scenario 1 after about 5 h corresponds to a tank serving a great number being emptied, thus corresponding in sudden change in served demand (slope). The relationship between served demand and curve slope is not so evident for failure scenario 2, since slope curve only relates to those users not directly served by the DWTP.

### 4.3 Sensitivity to tank levels

In case of power shutdown, the transient behaviour of the system is dictated by the amount of water stored in tanks. In order to better understand the relevance of each storage tank in the system, a sensitivity analysis has been performed. In particular, a sensitivity matrix is calculated by numerically computing the derivative of head of each node with respect on the level of each tank. By examining the resulting data, two types of tanks are identified, according to their altitude. On the one hand,



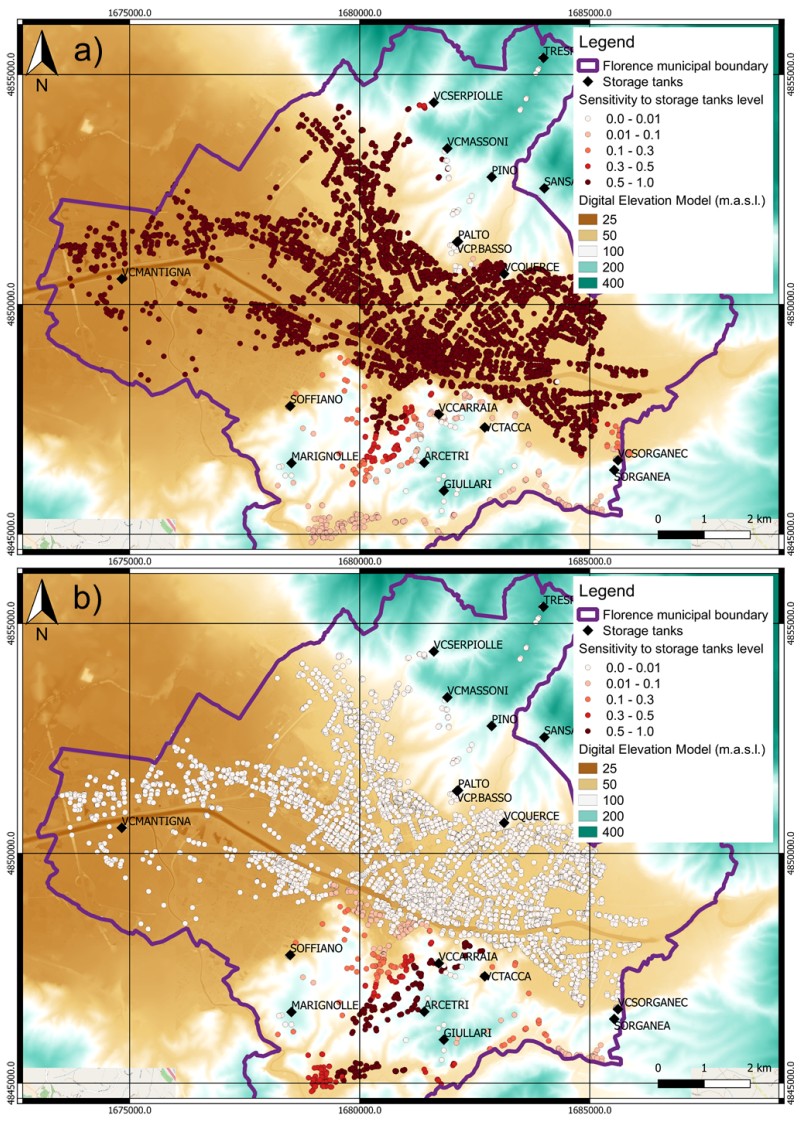

**Figure 10.** Digital Elevation Model of the study area and sensitivity to tank level for a lower tank (VC-Mantigna), panel a, left hand side) and for a upper tank (Arcetri), panel b, bottom). Reference coordinate system is EPSG:3003.

variations of water levels in low-altitude tanks strongly impact most network nodes, as shown in Figure 10 panel a, where the sensitivity to the "VCMantigna" storage tank is depicted. The nodes of the

network lying at elevations in the range 25–50 m – which represent the vast majority – undergo pressure variations of about 0.5–1 m for a 1-m variation of tank level. On the other hand, high-altitude



tanks, like for example the one labelled "Arcetri" shown in Figure 10 panel b), have a smaller area of influence limited to the immediate surroundings of the tank itself. This is also reflected by the longer service periods experienced by node pertaining to this areas, which share locally a relatively

abundant resource.

## 5 Conclusions

The impact of extreme weather events and natural disasters on urban structures and technological infrastructures, also in the perspective of climate change, is of increasing interest for citizens and institutions. In particular, the estimate of damage to network infrastructures, such as water supply

systems, poses an additional challenge due to the highly connected physical and functional topology, by which the detrimental effects spread to areas far from the event location, leading to indirect losses. In this work, a comprehensive methodology is defined to assess the impact of a flood on a WSS and implemented in an automated fashion. In particular, two main submodels are exploited: an *(i)* inundation model, which uses hydrometeorological data and a DTM to compute flood depth in the

case of an event with a given recurrence interval (hazard analysis), and a *(ii)* WSS hydraulic model, used to simulate fluid-dynamic behaviour of the network from topology, functional and demand data.

In the first place, a flood scenario is calculated by the inundation model, and water depths in the locations of WSS active components (pumps, electrically operated valve, etc.) are extracted. If water depth near an active component exceeds a given safety threshold, the component is considered failed

(exposure analysis) and its state is modified accordingly in the WSS model. Thereafter, the WSS model is run and nodal pressures are calculated. In this phase, users experiencing lack of service are identified as a function of time. Moreover, by comparing water pressure in the network with local flood depth, areas affected by backflow are identified. Finally, calculated data are aggregated to compute two time-varying metrics which quantify the global lack of service (through the number

of affected users) and global contamination extent (through the total length of pipes undergoing backflow).

The described method is applied to a case study. The studied area hosts 383 000 inhabitants on an area of $102 \, \text{km}^2$. The domestic water need – about $120\,000 \, \text{m}^3/\text{day}$ – is met by a WSS which abstracts the resource from the river Arno, flowing amidst the town. It is found that flood events with

a recurrence interval greater or equal to 100 years are those which affect functionality and safety of the WSS, namely by causing power disruption to the main lifting station. Two failure scenarios are defined and analysed, considering zero or one pump in operation respectively. Inundation maps of the area and service maps of the WDN are produced, thus identifying the most critical zones and the characteristic duration of service disruption in the two scenarios, and showing that providing a

backup system to keep only one pump in operation would largely reduce the affected population (by about 40%). As regards as the contamination of the pipeworks by floodwater, in the worst scenario



is is estimated that 68 to 100% of the network undergoes backflow risk depending on event duration, whereas the aforementioned improvement reduces length of pipeworks to be flushed by 60%, with a first-estimate saving of about 13 M€. Sensitivity of nodal pressures to tank levels is also studied, thus identifying influence areas of the various storage facilities.

The implemented methodology uses flood data, WSS topology and characteristics, and water demand data to compute WSS contamination risk maps and service maps at various time moments after the event. The model is automated and lightweight, the analysis being completed in few minutes, and can be effectively used in the strategic planning of disaster recovery procedures or in comparing network strengthening solutions in budget allocation activities.

Future developments may include studying the effect of first-intervention procedures (e.g. subsectioning of the network prior to the flood to select specific areas to contaminated while preserving functionality in others) and extending of the model to simulate recovery procedures, in order to estimate recovery times and transient network behaviour based on scheduling and available resources.

**Data availability**

**Author contribution**

First author conceived the impact assessment methodology and was responsible of flood hazard, exposure assessment, GIS operations and mapping. Second author implemented the PDD code, simulated the piping network and evaluated the impact metrics. Third author supervised the network modelling and last author promoted the research and supervised the flood risk aspects.

*Acknowledgements.* We acknowledge Publiacqua SpA for providing the sample network data and for the advisory given as stakeholder.

This research was supported financially by Fondazione Ente Cassa di Risparmio di Firenze under the research programme "ECRFI 2014".





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
