# Peer review of "Flood Impacts on a Water Distribution Network"

_Natural Hazards and Earth System Sciences, 2017_

## Referee Comment (RC1) · Anonymous Referee #1 · 24 Jul 2017

GENERAL COMMENTS

The manuscript proposes an operational methodology aimed at analysing both direct and indirect damages to a drinking water supply system due to a flooding event with a defined probability of occurrence. The methodology involves the combination of a flood model and an EPANET-based piping network model. The latter is developed with a pressure-driven approach, that allows to consider fully or partially non-operating nodes. The global impact of the flood on the network is evaluated considering both the number of inhabitants experiencing lack of services and the network damages due to pipe contamination.

The methodology is simple and allows to operationally evaluate the damages of a flood with a comprehensive approach, increasing the accuracy of the estimates considering also the indirect damages. The importance of taking into account the indirect costs

is clearly explained in the introduction of the manuscript and supported from the outcomes of the reported case study, that concerns the application of the methodology to the Water Supply System (WSS) of the city of Florence (Italy). The reported metrics, obtained considering inundation maps related to various return periods, are simple and adequate to demonstrate the significance of the damages to the WSS in the analysis of the flood-related hazard. The worst considered failure scenario leads to a percentage of affected population approaching 50% and an estimated cost of about 21 Mio €

The manuscript does not suggest new methodologies, but focuses on a relevant question related to the evaluation of the flood risk, exploiting consolidated instruments to provide a comprehensive operational framework. The combination of inundation maps and EPANET maps results in a useful operational tool for the assessment of the hazard related to the interaction between flood and WSS, fully compatible with the scope of NHESS. The manuscript is quite well-structured and all the steps of the procedure are clearly explained and easily replicable. Results are analysed in a deep and exhaustive way. Some minor suggestions related to the structure of the manuscript and the exposition of the results are pointed out in the "Specific comments" section.

Writing style and use of English should be improved. Sometimes unclear language structures hinder the readability of the manuscript. The work would benefit from extensive English editing by a native speaker.

SPECIFIC COMMENTS

1) Many flood models are available in the literature, amenable for different levels of complexity and different spatio-temporal resolutions (e.g., Fewtrell et al., 2011). The authors themselves state that inundation maps from local and national water authorities could be used, if available with an adequate spatial resolution (L 145-148). The inundation model adopted in the case study (Arrighi et al. 2013) is briefly described in section 2.1. Considering that the definition of the inundation map is a crucial step of the whole procedure and that the work aims at providing a complete and replicable tool

for the flood hazard assessment, the authors should explicit the reasons behind the choice of the inundation model. In general, they should provide some consideration on the applicability of this step to a generic case study (e.g., the proposed model can be applied as it independently from the local condition? Are there any framework in which different methodologies could be required? How can the "adequate spatial resolution" for the following steps of the procedure should be identified?)

2) The results section looks quite fragmented, with many short sub-sections and many separated figures. Some significant topics (e.g, the tank dynamics and the sensitivity to the tank levels) are just briefly introduced for the first time at the end of the section. The author should try to review the structure of the section trying to make it more fluent and readable, with some editorial improvements. E.g., the "Results" section would probably benefit from adding a short introduction describing the different analyzed aspects and trying to merge together some of the figures (e.g. figure 5 – 6 -7 – 8) using panels and subfigures.

3) The comparison with the results of the analysis carried out considering only the direct cost is crucial for explaining the scientific relevance of the proposed methodology and the importance of the problem. In the manuscript, it is limited to some lines (LL. 345-349) in the "Results" section. Even if a description of the direct damages can be found in Arrighi et al. (2016), as reported in line 349, due to the importance of the topic some more information and comparison should be provided in the manuscript (e.g., referring not only to the economic cost but also to the number of affected people, etc.).

REFERENCES

Arrighi, C., Brugioni, M., Castelli, F., Franceschini, S., & Mazzanti, B. (2016). Flood risk assessment in art cities: the exemplary case of Florence (Italy). Journal of Flood Risk Management.

Arrighi, C., Brugioni, M., Castelli, F., Franceschini, S., & Mazzanti, B. (2013). Urban micro-scale flood risk estimation with parsimonious hydraulic modelling and census

data. Natural hazards and earth system sciences, 13(5), 1375.

Fewtrell, T. J., Duncan, A., Sampson, C. C., Neal, J. C., & Bates, P. D. (2011). Benchmarking urban flood models of varying complexity and scale using high resolution terrestrial LiDAR data. Physics and Chemistry of the Earth, Parts A/B/C, 36(7), 281-291.

---

## Author Comment (AC1) · 25 Jul 2017

On behalf of my co-authors I would like to thank the referee #1 for his comments and suggestions. In the revision and polishing phases the manuscript will be thoroughly checked by an English native speaker to improve readability. I am now giving a short reply to the specific comments, which will be addressed in more detail in the revised manuscript.

1) The flood model here adopted and described in Arrighi et al. (2013) is very parsimonious from the numerical point of view. However, it has been demonstrated to be good enough to model water depths in the urban area when compared to historical watermarks. Thus, for the purpose of the study, an acceptable estimation of the maximum flood depth is considered as appropriate to make a control on the pressures of the nodes of the network. This implies to be able to capture the elevation of the road

network by using high resolution LiDAR-derived digital terrain models. The size of the study domain and the need of a very high mesh resolution, e.g. of the order of 0.5 m to represent the streets/building pattern, would imply a few million cells to be simulated in a full 2D model, with consequent specific hardware requirements. The use of the parsimonious flood model is however not suggested for flat topographies where full 2D models are recommended.

2) The results section will be better organized to improve its structure.

3) A more detailed comparison with the direct damages described in previous works will be included considering monetary losses, affected population and cultural heritage in order to better communicate the global flood risk context of the study area.

—————————————————

---

## Referee Comment (RC2) · Anonymous Referee #2 · 16 Aug 2017

L15: I would specify which the "worst failure scenario" is (features/return period). L36-39: considerable importance is given to "interdependency". Although this is a very interesting topic, the study is not really focused on interdependency (one hazard, one network). Indeed, I would reduce the text of the Introduction dedicated to interdependency, and just mention the relevance of this study in relation to the broader issue of cascading effects. FIG. 1: some arrows are not drawn. L241: in the abstract, the inhabitants were 385,000... L242: I would specify the name of the river (Arno). L247: rephrase "Flood risk in the area studied is estimated". L250: how about the societal costs? L274-276: 0.5m is the threshold above which failure is determines. How was this thresholds chosen? On which basis? This is very importance since all the results depend from this number. It is not specify how this thresholds was decided or obtained. This needs to be fixed, since an "arbitrary number" is not enough. FIG. 3: specify better what it is meant for "Flooded area" in the caption. Is the "flooded area"

[Figure]

identified by flood depth >= 0.5m? L279-283: is the average depth really significant for such area? I think it would be better to identify significant hotspots (points) in which the flood depths are measured for each scenario (and compared in order to get an idea of the event magnitude). TABLE 3: insert "0" instead of "-". L334-346: this paragraph should be moved to the Introduction, as it reports some literature. L349: does Arrighi et al. (2016) analyze the same city and the same scenario? L389: in the Conclusion, a "given safety threshold" is mentioned, that refers to the arbitrary number of 0.5m. As commented above, this threshold numbers should be justify, since it affects all the results. Or at least, a bit of discussion about it is needed. Could a shift from a binary consideration of flooding (>=0.5m – flood; < 0.5m – not flood) to a function (flood = function(water depth)) be a future progress of the study?

---

## Author Comment (AC2) · 26 Aug 2017

On behalf of my co-authors I would like to thank the referee #2 for his comments and suggestions. I am now giving a short reply to the specific comments, which will be addressed in detail in the revised manuscript.

1) The worst failure scenario is the 500 years recurrence interval and will be specified in the revised version of the abstract.

2) The interdependency is not only between the hazard and the network but also inside the network. In this sense, we strongly focused on interconnection. In fact the cascade effect on the pressure at nodes (i.e. the following possible need of decontamination) does not only depend on flood depth but also on the user demand, terrain morphology and network topology. Multiple interactions between WDN and flood are first in the triggering mechanisms and then in the spatial distribution of flood parameters.

[Figure]

3) Fig.1 will be checked.

4) The referee is right, this is a typo.

5) The name of Arno river will be specified.

6) L247 the sentence will be rephrased.

7) Societal costs have not been previously estimated, since out of the scope of the previous work.

8) The 0.5 m threshold has been identified based on the judgement of experts (network managers) who undertaken a 'what-if' analysis to evaluate the vulnerability of active components. This threshold has been considered as conservative with respect to the mean position of electric devices (e.g. control panels, sensors) observed in the plants. This clarification will be added.

9) Fig. 3 the flooded area represents the portion of territory where flood depth exceeds 0.01 m during the events. The caption will be modified.

10) The description of the flood scenarios will be improved to better characterize the events. Maximum flood depths in the historic and suburban districts reach 3.5 m and 4.5 m respectively for the 500 years recurrence interval (see fig. 4).

11) Table 3 will be modified.

12) The paragraph L334-346 will be moved to the introduction.

13) The work by Arrighi et al (2016) considered the 200 years flood scenario. Here the length of the contaminate pipeworks (e.g. the costs) for the 200 years and 500 years scenarios do not differ significantly. The sentence will be clarified.

14) A more detailed discussion will be added to the conclusion about the adopted threshold. A planned future work will consider a dynamic coupling of the flood and network model and in this case a function of the water depth will be adopted.

---

## Author Response (AR1)

**Responses to reviewers**

**Reviewer 1**

I would like to thank referee #1 for its fruitful comments, which helped improving the readability and quality of the manuscript. The text has been checked for typos and grammar mistakes, moreover some sentences have been rephrased to improve the readability.

You may find below a point by point reply to your comments and how they have been addressed in the revised manuscript.

| Comment | Reply |
|---|---|
| 1) Many flood models are available in the literature, amenable for different levels of complexity and different spatio-temporal resolutions (e.g., Fewtrell et al., 2011). The authors themselves state that inundation maps from local and national water authorities could be used, if available with an adequate spatial resolution (L 145-148). The inundation model adopted in the case study (Arrighi et al. 2013) is briefly described in section 2.1. Considering that the definition of the inundation map is a crucial step of the whole procedure and that the work aims at providing a complete and replicable tool for the flood hazard assessment, the authors should explicit the reasons behind the choice of the inundation model. In general, they should provide some consideration on the applicability of this step to a generic case study (e.g., the proposed model can be applied as it independently from the local condition? Are there any framework in which different methodologies could be required? How can the "adequate spatial resolution" for the following steps of the procedure should be identified?) | The flood model here adopted and described in Arrighi et al. (2013) is very parsimonious from the numerical point of view. However, it has been demonstrated to be good enough to model water depths in the urban area when compared to historical watermarks. Thus, for the purpose of the study, an acceptable estimation of the maximum flood depth is considered as appropriate to make a control on the pressures of the nodes of the network. This implies to be able to capture the elevation of the road network by using high resolution LiDAR-derived digital terrain models. The size of the study domain and the need of a very high mesh resolution, e.g. of the order of 0.5 m to represent the streets/building pattern, would imply a few million cells to be simulated in a full 2D model, with consequent specific hardware requirements. The use of the parsimonious flood model is however not suggested for flat topographies where full 2D models are recommended.

 The following paragraph has been added to section 2.1 to clarify the applicability of the method to a generic case study.
 "For the risk assessment of the WDS, exposure analysis is conducted on active components (Fig. 1) based on the maximum water depth occurring during the flood event. Maximum water depth is also used to assess the potential contamination at nodes. The selection of a suitable inundation model giving accurate flood depths depends on the characteristics of the domain, i.e. area, topography etc., although a spatial resolution of the order of 1 m (e.g. LiDAR derived products) should be preferred |

| | in urban areas to represent the streets/building pattern."

 Moreover, the suggested reference has been added (Fewtrell et al. 2011). |
|---|---|
| 2) The results section looks quite fragmented, with many short sub-sections and many separated figures. Some significant topics (e.g, the tank dynamics and the sensitivity to the tank levels) are just briefly introduced for the first time at the end of the section. The author should try to review the structure of the section trying to make it more fluent and readable, with some editorial improvements. E.g., the "Results" section would probably benefit from adding a short introduction describing the different analyzed aspects and trying to merge together some of the figures (e.g. figure 5 – 6 -7 – 8) using panels and subfigures | An introduction has been added to the results section to briefly presents its contents as follows "The results section is divided into three subsections. First, flood hazard scenarios are illustrated and the exposure analysis of the WSS components is described. Two failure scenarios with different residual functionality of the exposed lifting station are selected (sect. 4.1). Second, the dynamics of the WDS, i.e. temporal evolution of pressure at nodes and volume in the tanks are described. The population not served and contaminated pipe length defined as impact metrics are shown for the two failure scenarios (sect. 4.2). Third, the results of the sensitivity analysis of the WDS with respect to tank levels are presented (sect. 4.3). The role of the tank levels is crucial to satisfy the population demand during the transient after power shutdown and two different tank behaviours are mapped."

 The structure of the section has been modified reducing the number of subsections (from 6 to 3).

 Figures 5-6 and 7-8 have been merged into single figures with two panels. |
| 3) The comparison with the results of the analysis carried out considering only the direct cost is crucial for explaining the scientific relevance of the proposed methodology and the importance of the problem. In the manuscript, it is limited to some lines (LL. 345-349) in the "Results" section. Even if a description of the direct damages can be found in Arrighi et al. (2016), as reported in line 349, due to the importance of the topic some more information and comparison should be provided in the manuscript (e.g., referring not only to the economic cost but also to the number of affected people, etc.). | The referee raised an important point.
 The following paragraph has been added after the comments on the population not served in the two failure scenarios:
 "The population not served by the WDN exemplifies the large spread between direct and indirect flood impacts. In fact, for the 200 years and 500 years flood scenarios the residents inside the flooded area, i.e. directly affected, are ~35.6\% and ~44.8\% of the total population respectively. This means that if the indirect impact on the WDN is considered, the population affected by the flood almost doubles."
 Moreover, the concept is also recalled in the conclusive section as follows:
 "Although economic losses to the WDN, i.e. costs of decontamination of pipeworks, are almost negligible with respect to the direct losses to buildings, contents and artworks estimated in a previous work (Arrighi et al., 2016), the calculation of the population not |

| | served reveals that for a 200 years flood and worst failure scenario the population experiencing the lack of freshwater is almost three times the population directly flooded. This has crucial implications also on the post-emergency management and civil protection actions since interventions are required also outside the inundated area." |

**Reviewer 2**

I would like to thank referee #2 for its comments and suggestions, which helped improving the readability and quality of the manuscript. You may find below a point by point reply to your comments and how they have been addressed in the revised manuscript.

| Comment | Reply |
|---|---|
| L15: I would specify which the "worst failure scenario" is (features/return period). | The abstract has been corrected as follows "Results show that for the worst failure scenario (no residual functionality of the lifting station and 500 years flood) 420 km of pipeworks would require flushing and disinfection with an estimated cost of 21Mio euro, which is about 0.5% of the direct flood losses evaluated for buildings and contents." |
| L36-39: considerable importance is given to "interdependency". Although this is a very interesting topic, the study is not really focused on interdependency (one hazard, one network). Indeed, I would reduce the text of the Introduction dedicated to interdependency, and just mention the relevance of this study in relation to the broader issue of cascading effects. | The interdependency is not only between the hazard and the network but also inside the network. In this sense, we strongly focused on interconnection. In fact, the cascade effect on the pressure at nodes (i.e. the following possible need of decontamination) does not only depend on flood depth but also on the user demand, terrain morphology and network topology. Multiple interactions between WDN and flood are first in the triggering mechanisms and then in the spatial distribution of flood parameters which affects the risk of contamination. |
| FIG. 1: some arrows are not drawn. | The figure has been checked |
| L241: in the abstract, the inhabitants were 385,000 | The approximate number of inhabitants is 380000, the text has been checked to ensure consistency. |
| L242: I would specify the name of the river (Arno). | The name of the river has been specified. |
| L247: rephrase "Flood risk in the area studied is estimated". | The sentence has been rephrased |
| L250: how about the societal costs? | Societal costs have not been previously estimated, however a comparison between |

| | directly and indirectly affected people has been added to the result and conclusion sections. Indeed, the calculation of the population not served reveals that for a 200 years flood and worst failure scenario the population experiencing the lack of freshwater is almost three times the population directly flooded. This information has been added to the results section and it is also mentioned in the abstract. |
|---|---|
| L274-276: 0.5m is the threshold above which failure is determines. How was this thresholds chosen? On which basis? This is very importance since all the results depend from this number. It is not specify how this thresholds was decided or obtained. This needs to be fixed, since an "arbitrary number" is not enough. | The 0.5 m threshold has been identified based on the judgement of experts who undertook a 'what-if' analysis to evaluate the vulnerability of active components. This threshold has been considered as conservative with respect to the mean position of electric devices (e.g. control panels, sensors) observed in the plants. This clarification has been added to section 3.2. |
| FIG. 3: specify better what it is meant for "Flooded area" in the caption. Is the "flooded area" identified by flood depth >= 0.5m? | The flooded area represents the portion of territory where flood depth exceeds 0.01 m during the events. The caption has been modified. |
| TABLE 3: insert "0" instead of "-". | "-" has been replaced with "0" in Table 3. |
| L279-283: is the average depth really significant for such area? I think it would be better to identify significant hotspots (points) in which the flood depths are measured for each scenario (and compared in order to get an idea of the event magnitude). | In the first section of the results (now called 4.1 Flood and failure scenarios) a better description of the inundation characteristics has been added as follows. "Figure 3 shows the results of hazard analysis. For the 30-year RI an area of 2.5km$^2$ is flooded, with an average water depth of 1m. For this flood scenario two areas are affected by the flood, one upstream of the historic city in the right bank (right hand side of Fig.3) and one downstream on the left bank (left hand side of Fig.3. In the upstream area, flood depths are of the order of 0.3 m, while in the downstream area water depths reach locally 4 m in correspondence of excavation zones. For the 100-year recurrence interval, the flooded area increases to 12.7 km$^2$ with an average flood depth of about 1m. In the downstream areas water depth locally reaches up to 2.5 m in the right bank. For higher RI (200 and 500 year), the affected areas rise to 20 and 27 km$^2$ and average depths to 1.2 and 1.7 m respectively. In these scenarios also the historic district are affected (center of Fig.3 with water depths in the most depressed areas up to 4 m. |

| | In the downstream areas water depths are locally above 4 m (see Fig.4 for an example of 500-year RI inundation)." |
|---|---|
| L334-346: this paragraph should be moved to the Introduction, as it reports some literature. | The paragraph has been moved to the introduction as suggested by the referee. |
| L349: does Arrighi et al. (2016) analyze the same city and the same scenario? | The work by Arrighi et al (2016) considered the 200 year flood scenario. Here the length of the contaminate pipeworks (e.g. the costs) for the 200 year and 500 year scenarios do not differ significantly. The sentence has been clarified |
| L389: in the Conclusion, a "given safety threshold" is mentioned, that refers to the arbitrary number of 0.5m. As commented above, this threshold numbers should be justify, since it affects all the results. Or at least, a bit of discussion about it is needed. Could a shift from a binary consideration of flooding (>=0.5m – flood; < 0.5m – not flood) to a function (flood = function(water depth)) be a future progress of the study? | In section 3.2 the selection of the threshold has been clarified and the use of the safety threshold has been added to the conclusions. A planned future work will consider a dynamic coupling of the flood and network model and in this case a function of the water depth will be adopted. |

[revised manuscript text omitted]